# A Sponge-like Polysaccharide from Pine Pollen: Structural Features and Therapeutic Potential in DNCB-Induced Atopic Dermatitis Models

**DOI:** 10.3390/pharmaceutics17081058

**Published:** 2025-08-14

**Authors:** Zhuoya Qiu, Mengru Zhang, Haitao Du, Yi Wang, Xuekun Shao, Jialei Fu, Ping Wang, Cheng Wang

**Affiliations:** 1College of Pharmacy, Shandong University of Traditional Chinese Medicine, Jinan 250014, China; qiuzhuoya@163.com (Z.Q.); sxk18271888@163.com (X.S.); 2Institute of Traditional Chinese Medicine Pharmacology, Shandong Academy of Chinese Medicine, Jinan 250014, China; 15625156296@163.com (M.Z.); kkitdht@foxmail.com (H.D.); wyi_1989@163.com (Y.W.); jialf-007@163.com (J.F.)

**Keywords:** pine pollen, polysaccharides, atopic dermatitis, epidermal injury

## Abstract

**Objectives:** Atopic dermatitis (AD) is a long-term, recurring inflammatory skin condition characterized by impaired epidermal barrier function and abnormal immune system regulation. Pine pollen has traditionally been used for dermatological treatments, though its active components remain unclear. The primary objective of this study was to pinpoint the active constituents of pine pollen and elucidate its therapeutic effects against AD. **Methods:** The safety concentration ranges and protective efficacy of nine pine pollen constituents against 2,4-dinitrochlorobenzene (DNCB)-induced HaCaT cell damage were evaluated using the CCK-8 assay. Furthermore, models of DNCB-induced damage were established both in vitro (HaCaT cells) and in vivo (BALB/c mice) to explore the protective effects of the key functional component. **Results:** Our findings identified pine pollen polysaccharides (PPPS) as the principal bioactive constituent, characterized by a unique infrared absorption spectral profile and a sponge-like architecture with three-dimensional interconnected porous networks. In vitro, PPPS inhibited DNCB-induced decreases in cell viability, morphological abnormalities, oxidative stress, and apoptosis. In vivo, PPPS alleviated DNCB-induced skin lesions by attenuating epidermal hyperplasia, suppressing mast cell infiltration, inhibiting cell apoptosis, and downregulating the expression of IL-4 and IL-17A. **Conclusions:** This study provides evidence that PPPS from pine pollen can alleviate epidermal damage in AD, offering a novel therapeutic strategy for AD treatment.

## 1. Introduction

Atopic dermatitis (AD), one of the most prevalent dermatological disorders, is characterized by eczematous inflammatory skin lesions and severe pruritus, affecting approximately 10% of adults and 20% of children worldwide [1]. Strikingly, AD affects 30.48% of Chinese infants (1–12 months), with this proportion continuing to rise annually [2,3]. Previous studies indicate that AD pathogenesis arises from genetic predisposition, environmental triggers, and autoimmune dysregulation, with Th immune imbalance being the pivotal pathogenic mechanism. However, recent studies have revealed that damage to keratinocytes, which constitute approximately 90% of epidermal cells, serves as a pivotal factor in the development of AD [4,5,6]. AD is currently treated using oral and topical immunomodulators, such as corticosteroids and cyclosporine A, but prolonged use can lead to significant adverse effects [7,8].

*Pinus massoniana* Lamb. (phylum Gymnospermae, order Pinales, family Pinaceae) is a globally distributed coniferous species [9]. Pine pollen is obtained through the collection of the microstrobili (male cones), followed by desiccation and impurity removal [10]. The application of pine pollen in traditional Chinese medicine dates back to the Qing Dynasty, with its earliest documented use for dermatological conditions appearing in the Ben Cao Cong Xin during the Qing Dynasty. The Pharmacopoeia of the People’s Republic of China (ChP 2020) also recognizes its topical efficacy in desiccation and promoting hemostasis, with clinical indications including eczema, pyoderma, traumatic hemorrhage, cutaneous erosion, and diaper dermatitis. Furthermore, clinical evidence has also demonstrated the significant therapeutic and prophylactic efficacy of pine pollen in the treatment of various skin disorders [10,11,12]. However, the specific active constituents of pine pollen and its potential mechanisms of action against AD remain to be elucidated.

Accordingly, the primary active constituents of pine pollen were identified through systematic pharmacological evaluation in this study, followed by in vivo and in vitro experimentation to elucidate the therapeutic effects.

## 2. Materials and Methods

### 2.1. Materials and Reagents

Macklin Biochemical Technology Co., Ltd. (Shanghai, China) supplied the following compounds: p-Hydroxybenzoic acid (PHB), Protocatechuic acid (PCA), 4-Hydroxy benzaldehyde (HBA), Isovanillic acid (IVA), Succinic acid (SA), Kaempferol (KAE), Glyceryl monostearate [(2S)-2,3-dihydroxypropyl octadecenoate] (GMO), and β-sitosterol (sitosterol). Pine pollen polysaccharides (PPPS) were extracted using a previously described method [13,14]. In total, 100 g of broken-wall pine pollen was defatted via reflux extraction with petroleum ether at 50 °C for 3 h and subsequently extracted via reflux with 95% ethanol at 70 °C for 2 h to remove impurities; the resulting material was then subjected to aqueous extraction by boiling it with 8-fold (*w*/*w*) water addition three times. After concentration of the combined filtrates to 200 mL, the polysaccharide solution was subjected to deproteinization with Sevage reagent (chloroform:butanol, 4:1 *v*/*v*) in a 3:1 volume ratio. The polysaccharide solution layer was collected, absolute ethanol was added to a final concentration of 80%, the precipitate was collected via centrifugation at 3000× *g* for 10 min, and the final product (PPPS) was obtained through lyophilization. The following monosaccharides and uronic acids were all procured from Sigma-Aldrich (St. Louis, MO, USA): Rhamnose (Rha), Fructose (Fru), Galacturonic Acid (Gal-UA), Xylose (Xyl), Mannose (Man), Glucuronic Acid (Glc-UA), Arabinose (Ara), Ribose (Rib), Fucose (Fuc), Galactose (Gal), Mannuronic Acid (Man-UA), Glucose (Glc), Guluronic Acid (Gul-UA). 2,4-dinitrochlorobenzene (DNCB) was purchased from Aladdin Ltd. (Shanghai, China). Dexamethasone and Hoechst 33342/PI Double Stain Kits were sourced from Solarbio Science & Technology Co., Ltd. (Beijing, China). Annexin V-FITC/PI Apoptosis Detection Kits were procured from Vazyme Biotechnology Co., Ltd. (Nanjing, China). Beyotime Biotechnology Co., Ltd. (Shanghai, China) provided the Cell Counting Kit-8 (CCK-8) and Reactive Oxygen Species (ROS) Assay Kit. Hematoxylin and eosin (H&E), Fluorescein Tunel Cell Apoptosis Detection Kit, 0.5% toluidine blue, and 3,3′-diaminobenzidine (DAB) were supplied by Servicebio Co., Ltd. (Wuhan, China). Anti-IL-4 (AF5142) and anti-IL-17A (DF6127) antibodies were commercially sourced from Affinity Biosciences (Changzhou, China).

### 2.2. Bioactive Constituents of Pine Pollen

Chemical analysis based on the Traditional Chinese Medicine Systems Pharmacology Database (TCMSP) identified 8 principal constituents in pine pollen (as of 1 January 2025): PHB, PCA, HBA, IVA, SA, KAE, GMO, and sitosterol (Figure 1) [15]. Polysaccharides have recently been proven to be effective active ingredients of pine pollen [14]. A PPPS is a kind of polar, hydrophilic macromolecule, such as glucose, galactose, arabinose, glucuronic acid, and others, connected via a glycosidic bond of monosaccharide composition. PPPS are branched polymers. The molecular weight of PPPS, assessed using various extraction methods, ranges between 25 and 128 kDa [13]. Based on previous studies confirming the therapeutic effect of PPPS on skin injuries and considering their anti-inflammatory properties [14], we incorporated PPPS into this investigation. Consequently, this study examines the bioactivities of 9 pine pollen components, comprising the 8 previously characterized small molecules and PPPS.

### 2.3. Cell Culture and Treatment

HaCaT cells (iCell, Shanghai, China) were maintained in DMEM (VivaCell, Shanghai, China) containing 10% FBS (ExCell, Suzhou, China) and 1% penicillin–streptomycin (Beyotime, Shanghai, China) in a humidified 37 °C/5% CO_2_ incubator.

Following seeding in 96-well plates (density: 1 × 10^4^ cells/well), HaCaT cells were maintained until achieving 80–90% confluence. DNCB was used to establish the HaCaT cell injury model [16,17]. Cells were then treated with 9 pine pollen components and/or DNCB, with all agents dissolved in the culture medium containing 2% FBS. For the determination of safe concentrations of active components, cells were exposed to various concentrations of pine pollen components for 24 h. Dose optimization for DNCB modeling involved 24 h exposure to graded concentrations (2.5, 5, 10, 20, and 40 μM). For the screening of effective components, cells were co-treated with DNCB (at the modeling concentration) and pine pollen components (at safe concentrations) for 24 h before subsequent experiments.

### 2.4. Cell Viability and Toxicity Measurements

Post-treatment viability was quantified via CCK-8 assay. Following medium aspiration, cells were incubated at 37 °C for 0.5–2 h with 100 μL of CCK-8 reagent (DMEM: CCK-8 = 9:1), with subsequent OD_450_ nm measurement.

### 2.5. FTIR, Monosaccharide Composition, and SEM Detection of PPPS

Fourier transform infrared spectroscopy (FTIR): PPPS powder (2 mg) was thoroughly mixed with KBr (150 mg) and pressed into 1 mm tablets for FTIR analysis. Spectral absorption (4000–400 cm^−1^) was acquired using a Nicolet Is5 FT-IR spectrometer (Thermo Fisher, Walthamm, MA, USA). Monosaccharide profiling of PPPS (5.0 mg) involved hydrolysis in 2 M TFA (2–3 mL, 121 °C, 2 h). After nitrogen drying and methanol washing, the hydrolysates were reconstituted in deionized water and filtered (0.22 μm pore size). Subsequent quantification was performed via high-performance anion-exchange chromatography (HPAEC) using a CarboPac PA-20 analytical column (3 × 150 mm). The following gradients were employed during elution (0.5 mL/min): A (H_2_O), B (0.1 M or 250 mM NaOH), and C (0.2 M NaAc/0.1 M NaOH or 1 M NaAc). Finally, identification of monosaccharides was achieved through retention time and peak area comparisons with reference standards. In the scanning electron microscopy (SEM) analysis of PPPS (Zeiss Merlin Compact, Oberkochen, Germany), we utilized gold-sputtered specimens mounted on substrates, imaged at 1.0 kV with 5000× and 10,000× magnifications.

### 2.6. PI Staining

To assess the damage dealt to the DNCB-induced HaCaT cells treated with PPPS, PI/Hoechst staining was performed. The cells underwent triple PBS washing, followed by incubation in 500 μL staining buffer supplemented with PI (5 μL) and Hoechst (5 μL). Post-mixing, the cells were incubated (4 °C, 30 min) and then subjected to a final PBS wash. Fluorescent images were captured using an inverted fluorescence microscope (Nikon, Tokyo, Japan). All images were analyzed with ImageJ to quantify PI-positive (dead/damaged) and Hoechst-positive (total) cells. Specifically, images were imported into ImageJ and converted to 8-bit format. Following thresholding and binarization, sequential processing with Fill Holes (to close intracellular gaps) and Watershed (to separate adherent cells) was applied. The minimum area threshold for the Analyze Particles function was determined by measuring a population of representative small cells. Cell counts were quantified using this size criterion. Manual verification was performed on 10% of randomly selected images (if discrepancies were >5%, manual counting was conducted using Image-Pro Plus 6.0’s Multi-Point tool).

### 2.7. ROS Detection

Reactive oxygen species (ROS) assays were used to measure intracellular levels according to kit instructions. Briefly, cells from different treatment groups were 2,7-Dichlorodihydrofluorescein diacetate (DCFH-DA)-stained (10 μM) and incubated (37 °C, 20 min). After a wash with a serum-free medium, fluorescence images were acquired using an inverted fluorescence microscope. Additionally, stained cells were collected, and the mean FITC fluorescence intensity was immediately analyzed via flow cytometry.

### 2.8. Flow Cytometry of Apoptosis

Apoptosis was quantified in HaCaT cells via flow cytometry with an Annexin V-FITC/PI apoptosis detection kit (Nanjing, China). After 24 h of treatment, cells were harvested, washed twice with ice-cold PBS, and resuspended in 0.1 mL of 1× binding buffer containing 5 μL PI and 5 μL Annexin V-FITC. After 10 min of incubation in the dark (RT), 0.4 mL of 1× binding buffer was added. Data acquisition was performed using a BD Biosciences flow cytometer (San Jose, CA, USA), with subsequent analysis in FlowJo software (v7.6). Annexin V-FITC-positive/PI-negative cells were classified as early-apoptotic (lower-right quadrant—LR), while dual-stained cells were considered necrotic/late-apoptotic (upper-right quadrant—UR).

### 2.9. Animal Experiments

Male BALB/c mice (6 weeks old) were acquired from Jinan Pengyue Experimental Animal Breeding Co., Ltd. (Jinan, China). The Animal Ethics Committee of the Shandong Academy of Chinese Medicine approved all protocols (License No. SDZYY20240304004), which rigorously adhered to the Guide for the Care and Use of Laboratory Animals.

After a 1-week acclimatization period, 25 mice were randomly assigned to five groups (*n* = 5)—control, model, 2% PPPS, 5% PPPS, and Dexamethasone. Dorsal hair was removed under anesthesia one day prior to the experiment. For sensitization, 100 μL of 1% (*w*/*v*) DNCB, dissolved in acetone/olive oil in a 1:3 ratio, was administered topically to the shaved dorsal skin of the animals on days 1, 4, and 7 in the model and treatment groups. After a 1-week interval, 0.5% DNCB was applied every 3 days for 2 weeks to induce AD symptoms [18,19]. From day 14 onward, the treatment groups received daily topical applications of 2% PPPS, 5% PPPS (vaseline-based), or Dexamethasone, while the control and model groups received vaseline only [20,21]. On day 30, we carried out regrown hair removal followed by saline rinsing to eliminate residual depilatory cream. Following anesthesia, mice were euthanized via cervical dislocation. Dorsal skin tissues were then harvested and immersion-fixed in 4% paraformaldehyde.

### 2.10. Histopathological Analysis

To assess epidermal thickness and inflammatory infiltration, isolated dorsal skin samples from the mice, after fixation with 4% paraformaldehyde, were paraffin-embedded and sectioned at a thickness of 4 μm. Sections underwent deparaffinization in xylene followed by rehydration through an ethanol gradient (100%, 95%, 70%), before being stained with H&E for histopathological analysis. Images were acquired using a microscope (Nikon, Tokyo, Japan).

### 2.11. TUNEL Staining

Deparaffinized tissue sections were washed with PBS, incubated with 100 μL equilibration buffer at 25 °C for 20 min, and treated with 100 μL TdT solution (37 °C, dark, 60 min). After three PBS washes, nuclei were DAPI-counterstained and imaged using a fluorescence microscope (Nikon, Tokyo, Japan). Then, images were imported into ImageJ and converted to 8-bit format. The Multi-Point tool was used to count cells by clicking on individual cells.

### 2.12. Mast Cell Staining

After 15 min RT incubation in 0.5% toluidine blue, the dorsal skin sections underwent triple PBS washing. Mast cells (MCs) were dyed purple. Images were acquired using a light microscope (Nikon, Tokyo, Japan). Then, images were imported into ImageJ and converted to 8-bit format. The Multi-Point tool was used to count cells by clicking on individual cells.

### 2.13. Immunohistochemistry

Deparaffinized skin tissue sections were rinsed with PBS and subjected to antigen retrieval by immersion in citrate buffer (0.01 M, pH 6.0) at 95 °C for 20 min. For peroxidase blockade, we used 3% H_2_O_2_, and 5% bovine serum albumin (BSA) was employed for nonspecific binding inhibition. The sections were then incubated overnight at 4 °C with primary antibodies, including IL-4 (1:100, #AF5142, Affinity) and IL-17A (1:100, #DF6127, Affinity), and then the corresponding horseradish peroxidase (HRP)-conjugated secondary antibodies were applied and incubated at 37 °C for 1 h. DAB staining was followed by hematoxylin nuclear counterstaining. Microscopic examination was conducted for evaluation.

### 2.14. Statistical Analysis

Data are presented as means ± SD (*n* ≥ 3). SPSS 26.0 (IBM, Armonk, NY, USA) was used for one-way analysis of variance (ANOVA) and subsequent least significant difference (LSD) post hoc comparisons. GraphPad Prism 8.0 (GraphPad Software, San Diego, CA, USA) was used for data visualization and calculations. The results were deemed statistically significant if *p* < 0.05.

## 3. Results

### 3.1. Active/Toxic Effects of Pine Pollen Components

The CCK-8 assay revealed that PHB, PCA, HBA, SA, and GMO exhibited no significant effect on HaCaT cell viability at the tested concentrations (Figure 2). In contrast, IVA (80 μM), sitosterol (40 μM), and PPPS (100, 200, 400, and 800 μg/mL) demonstrated dose-dependent proliferative effects, with PPPS showing the most pronounced enhancement at 400 μg/mL. Conversely, KAE dose-dependently reduced HaCaT cell viability.

### 3.2. Screening of Pine Pollen Bioactive Components Alleviating Keratinocyte Damage

Based on established methodologies, a DNCB-induced keratinocyte injury model for AD was developed. CCK-8 analysis revealed that 24 h of DNCB exposure dose-dependently reduced HaCaT cell viability, with 10 μM identified as the optimal modeling concentration for subsequent experiments (Figure 3A).

To investigate the protective effects of pine pollen components against DNCB-induced cytotoxicity, HaCaT cells were co-treated with DNCB (10 μM) and the following components for 24 h: PHB (40 μM), PCA (40 μM), HBA (40 μM), IVA (80 μM), SA (40 μM), KAE (20 μM), GMO (40 μM), sitosterol (40 μM), or PPPS (400 μg/mL). The results revealed that KAE exacerbated the DNCB-induced suppression of cell viability. Conversely, SA and PPPS alleviated the decline, although the therapeutic effect of the former was not statistically significant (Figure 3B). These findings identify PPPS as the principal bioactive constituent counteracting DNCB-induced keratinocyte damage.

### 3.3. Structure of PPPS

The FTIR spectra of the PPPS are shown in Figure 4A. The signal at 3410 cm^−1^ was attributed to the stretching vibration of O–H groups. The absorption peak at 2931 cm^−1^ corresponds to C–H stretching, a characteristic feature of polysaccharides. Asymmetric and symmetric vibrations of the COO structure were observed at 1631 cm^−1^ and 1401 cm^−1^, respectively, indicating the presence of galacturonic acid. The absorption bands at 1241 cm^−1^ and 1042 cm^−1^ were assigned to O–H vibrations, with the strong absorbance at 1042 cm^−1^ being a distinctive marker of β-glycosidic bonds. Additionally, the weak absorption peak at 762 cm^−1^ is consistent with the skeleton bending of the pyranose ring. In Figure 4B, the results regarding monosaccharide composition show that PPPS were heteropolysaccharides composed of Rha (6.33%), Ara (27.01%), Gal (6.92%), Glc (45.72%), Xyl (5.83%), and Gal-UA (8.19%). The SEM results demonstrate that the surfaces of the PPPS exhibit a rough texture (Figure 4C). Further magnification reveals a three-dimensional network-like porous structure resembling a sponge, suggesting that PPPS may have excellent moisture retention capacity and biocompatibility properties.

### 3.4. PPPS Alleviate DNCB-Induced Keratinocyte Damage

The CCK-8 analysis revealed that PPPS dose-dependently inhibited the DNCB-induced reduction in HaCaT cell viability, with significant therapeutic effects observed at 200 and 400 μg/mL (Figure 5A). PI staining, which causes damaged/dead cells to emit red fluorescence, was employed to assess PPPS-mediated cyto-protection via fluorescence microscopy. DNCB significantly increased PI-positive cell counts, an effect unaltered by 100 μg/mL PPPS but significantly reduced at concentrations of 200 and 400 μg/mL (Figure 5B,C). These results demonstrate that effective doses of PPPS attenuate DNCB-induced cytotoxicity and cell death.

### 3.5. PPPS Inhibit ROS Overproduction

Oxidative stress, driven by excessive ROS generation, is a critical contributor to epidermal cell damage [22]. The results of fluorescence imaging show that DNCB treatment significantly increased ROS-positive cell counts, while PPPS dose-dependently reversed this effect (Figure 6A). Flow cytometry analysis corroborated these findings, showing an elevated mean FITC intensity in the DNCB-treated cells and significant reductions with PPPS (400 μg/mL) co-treatment (Figure 6B). These data indicate that PPPS suppress DNCB-induced ROS generation and subsequent oxidative damage.

### 3.6. PPPS Inhibit DNCB-Induced Apoptosis

Keratinocyte apoptosis has been validated as a critical pathological process in skin damage in AD [23,24]. Analysis via flow cytometry demonstrated a marked rise in the count of apoptotic cells within the DNCB-treated group. PPPS administration led to a dose-dependent reduction in apoptosis, with 200 and 400 μg/mL concentrations showing more pronounced therapeutic effects compared to lower doses (Figure 7). These results establish an inhibitory role for PPPS in DNCB-induced apoptosis in HaCaT cells.

### 3.7. PPPS Alleviate Dermal Injury in a DNCB-Induced AD-like Mouse Model

H&E staining revealed increased cellularity, epidermal hyperplasia, and inflammatory cell infiltration in a DNCB-induced AD-like mouse model, all of which were reversed by PPPS treatment. Both 2% and 5% PPPS administration significantly reduced epidermal thickness and inflammatory cell accumulation in a dose-dependent manner, with 5% PPPS achieving an efficacy comparable to that of Dexamethasone (Figure 8A,B). DNCB treatment significantly increased apoptotic cell counts in AD skin tissues. Both PPPS and Dexamethasone administration effectively suppressed this apoptotic response, particularly attenuating epidermal cell apoptosis (Figure 8C,D). These data demonstrate that the alleviation of AD-like skin lesions by PPPS occurs through dose-dependent dual mechanisms involving anti-inflammatory and anti-apoptotic pathways.

### 3.8. PPPS Reduce MC and Th Cell Infiltration in AD Skin

MCs serve as the principal effector cells mediating allergic responses in AD through the release of inflammatory mediators. Toluidine blue staining demonstrated significantly increased levels of MCs in the skin of the model mice, levels that were reduced by PPPS treatment (Figure 9A). The IHC analysis of Th2/Th17-associated cytokines revealed elevated IL-4 expression (Th2 biomarker) localized predominantly in the epidermal layer of the DNCB-induced AD-like mouse model, with PPPS administration significantly reducing IL-4 distribution, though in a less-pronounced manner than that of Dexamethasone (Figure 9B). Similarly, IL-17A (Th17 biomarker) exhibited heightened expression in both the epidermal and dermal layers of AD skin, which was suppressed by PPPS treatment (Figure 9C). Collectively, these findings demonstrate PPPS inhibit DNCB-induced MC infiltration and Th2/Th17 activation in AD skin.

## 4. Discussion

This study demonstrates that PPPS can alleviate HaCaT cell damage, inhibit ROS overproduction, reduce cell apoptosis, and regulate immunity across complementary AD models (in vivo and in vitro), revealing the potential way in which PPPS relieve AD symptoms.

Epidermal damage is a critical factor in assessing AD, and keratinocytes play a significant role in the skin wound-healing process [25,26,27]. The etiopathogenesis of AD is widely believed to be driven by the complex interplay between primordial and adaptive keratinocytes, immune cells, and signaling molecules such as interleukins IL-10, IL-13, interferons, and other cytokines [28,29]. Keratinocyte injury [30], abnormal proliferation, and differentiation [31] can further lead to skin barrier disruption, promoting the onset and progression of AD. Our findings indicate that PPPS exerts a protective effect on epidermal tissue by reducing keratinocyte damage, alleviating epidermal hyperkeratosis, and inhibiting apoptosis. AD symptom relief by fucoidan was mechanistically confirmed to involve the reduced mRNA expression of TARC, MDC, and RANTES in IFN-γ/TNF-α-exposed keratinocytes, decreased epidermal thickening, and inhibited mast cell infiltration [32]. Additionally, myricetin alleviates AD by improving skin barrier damage, inhibiting apoptosis, and reducing inflammatory responses [33]. These results underscore the significant role of PPPS in modulating these key indicators.

There is clear evidence that excessive production of ROS can lead to cellular apoptosis and tissue damage and contribute to chronic inflammation in neurodegenerative, cardiovascular, and metabolic diseases [34,35,36]. ROS can promote the initiation and development of AD via oxidative stress, leading to the secretion of T-cell differentiation, pro-inflammatory cytokines, and the worsening of skin symptoms [37]. In this research, we observed that DNCB treatment caused a marked rise in intracellular ROS levels, while PPPS treatment counteracted this effect in a dose-dependent way. By reducing ROS generation, PPPS can mitigate oxidative stress-induced damage to keratinocytes. This antioxidative stress effect of PPPS may be attributed to their ROS-neutralizing capability of antioxidant enzymes such as catalase (CAT) and superoxide dismutase (SOD) or to directly scavenge free radicals [38,39,40,41]. Previous studies have also demonstrated that many natural products with antioxidant properties can effectively alleviate AD symptoms by modulating the redox balance within the skin [42,43,44]. PPPS may represent a potential candidate in this aspect, though additional research is required to elucidate their specific molecular mechanisms in regulating the antioxidant system.

Immune dysregulation constitutes the fundamental pathogenic mechanism of AD, involving the activation and imbalance of numerous immune cells. The inhibition of the Th2/Th17 signaling pathways significantly alleviates AD [45,46,47]. Chen et al. demonstrated that phellodendrine ameliorates AD skin inflammation in mice by suppressing IL-4-induced STAT3 activation in keratinocytes [48]. MCs contribute to the early pathological process of AD via the MRGPRX2-mediated release of tryptase, which promotes the release of Th2 cytokines, thereby driving allergic reactions and pruritus [49]. Additionally, the function of Th17 cells, which secrete IL-17A, a pro-inflammatory mediator, cannot be overlooked. Reportedly, allicin directly inhibits the IL-17A/F-induced STAT3/NF-κB and TRAF6/MAPK/NF-κB signaling cascades in keratinocytes, thereby attenuating pro-inflammatory feedback and ameliorating psoriasis-like dermatitis [50]. In our study, it was observed that the expression of Th2 and Th17 cells was influenced by DNCB induction. PPPS treatment significantly reduced the number of MCs in the dermal tissue of the DNCB-induced AD-like mouse model and markedly reversed the increased expression of IL-4 and IL-17A. These findings align with prior evidence relating to the involvement of immune cells in the pathogenesis of AD.

Plant polysaccharides demonstrate therapeutic efficacy against atopic dermatitis (AD), though their mechanisms remain incompletely characterized. Zhang et al. revealed that honeysuckle polysaccharide (WLJP-025p) exerts anti-AD effects through the modulation of the MAPK/NF-κB/AP-1 axis [51]. Similarly, Liao et al. confirmed that Dendrobium officinale polysaccharide ameliorates DNFB-induced murine AD by suppressing MAPK/NF-κB/STAT3 signaling [52]. Our preliminary research established that PPPS alleviate LPS-induced myocardial inflammation by upregulating p110β expression and inhibiting PI3K/AKT/NF-κB activation [53]. This evidence collectively suggests the NF-κB pathway represents a pivotal mechanism for PPPS-mediated anti-AD activity. As the central transcriptional regulator for multiple pro-inflammatory genes, NF-κB critically governs immune and inflammatory responses [54], with its canonical pathway being crucial for effector T-cell differentiation and memory T-cell recall responses [55]. The disruption of canonical NF-κB signaling consequently impedes Th2/Th17 cell differentiation, downregulating the transcription of pro-inflammatory mediators including TNF-α, IL-17A, and IL-4 [56,57], thereby attenuating keratinocyte damage and immune dysregulation. Experimentally, PPPS significantly inhibited the DNCB-induced overexpression of IL-4 and IL-17A. Concurrently, its antioxidant properties likely indirectly dampen oxidative stress-mediated NF-κB activation, synergistically inhibiting inflammatory cascades and providing crucial mechanistic insights for the PPPS-mediated attenuation of AD-like symptoms.

While this study demonstrates that PPPS alleviate DNCB-induced AD-like damage, whether their therapeutic effect results from direct binding between PPPS and DNCB remains unclear. DNCB contains highly polar -NO_2_ and -Cl groups and functions as a hapten, forming immunogenic complexes through covalent binding with epidermal proteins [58]. Conversely, PPPS are a type of polysaccharide polymer rich in hydroxyl (-OH) and carboxyl (-COOH) groups [59]. Chemically, the potential for direct PPPS–DNCB interaction appears limited due to the insufficient structural basis for strong intermolecular forces. As a hapten, DNCB primarily mediates effects through specific protein binding, while PPPS—composed of multiple monosaccharides linked via β-glycosidic bonds—exhibits molecular polar groups with a greater propensity to form hydrogen bonds with aqueous-phase water molecules than to preferentially interact with DNCB’s nitro groups. Furthermore, in published polysaccharide intervention studies using DNCB-induced AD models [60,61,62,63], polysaccharide–DNCB binding remains uninvestigated, with therapeutic efficacy consistently being attributed to immunomodulatory pathways or antioxidant activity rather than a reduction in bioactive DNCB concentrations. This collective evidence suggests that polysaccharides exert therapeutic effects primarily by suppressing DNCB-triggered immune activation rather than through direct molecular interaction. Consequently, our findings more strongly support the fact that PPPS alleviate DNCB-induced AD-like symptoms through intrinsic bioactivity. Future studies employing surface plasmon resonance (SPR) and high-performance liquid chromatography (HPLC) could quantitatively assess potential interactions between polysaccharides and DNCB.

## 5. Conclusions

In summary, this study identifies PPPS as the crucial component of pine pollen for AD intervention. PPPS alleviate HaCaT cell injury, inhibit ROS generation, and suppress apoptosis in vitro. Additionally, PPPS administration ameliorated skin lesions, apoptosis, and inflammatory infiltration in a DNCB-induced AD-like mouse model. While PPPS exhibit therapeutic potential for the management of AD, additional mechanistic investigations are warranted to delineate their molecular targets and signaling pathways.

## Figures and Tables

**Figure 1 pharmaceutics-17-01058-f001:**
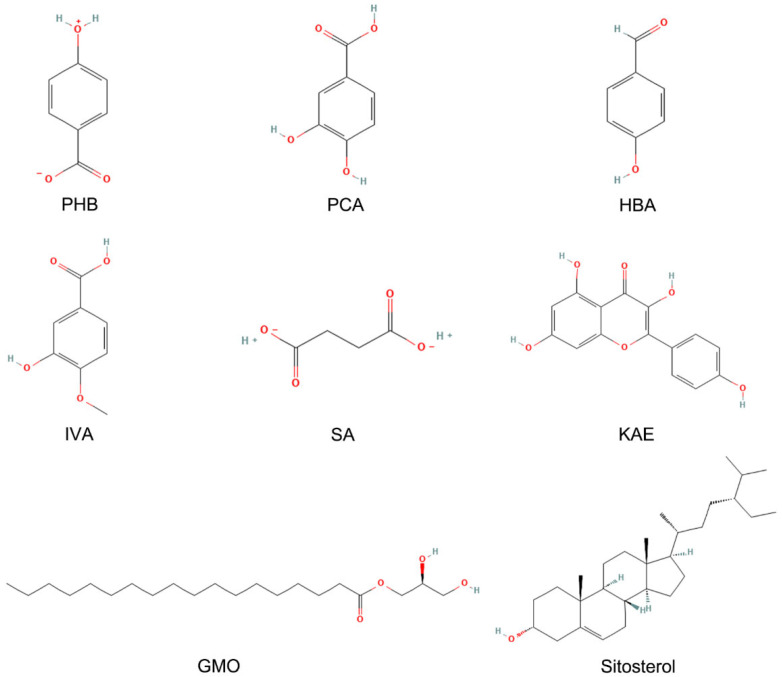
Composition of pine pollen in the TCMSP database.

**Figure 2 pharmaceutics-17-01058-f002:**
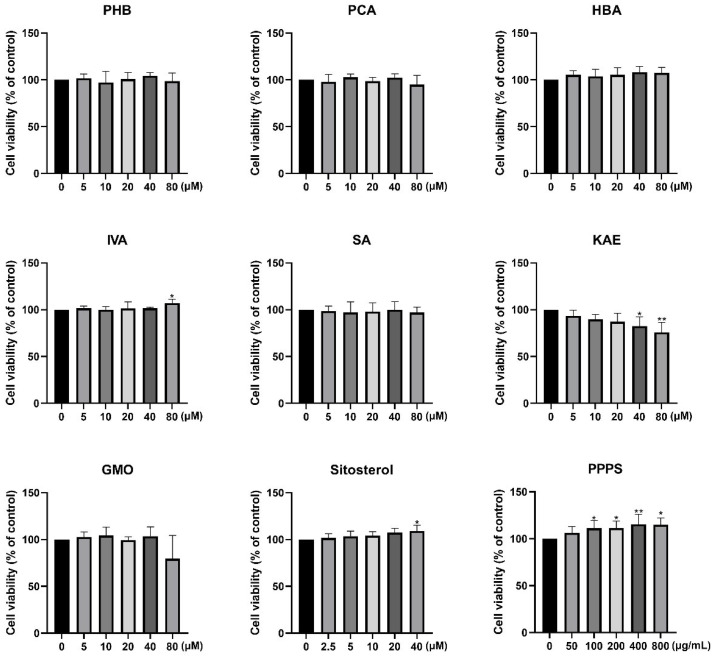
Effects of pine pollen components on HaCaT cell viability. * *p* < 0.05; ** *p* < 0.01.

**Figure 3 pharmaceutics-17-01058-f003:**
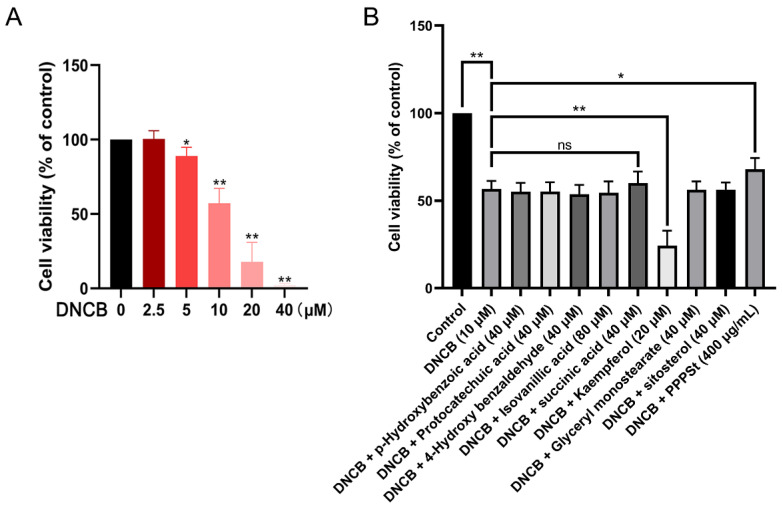
Effects of pine pollen components on DNCB-inhibited HaCaT cell viability. (**A**) Dose–response relationship of DNCB toxicity. (**B**) Therapeutic efficacy screening. * *p* < 0.05; ** *p* < 0.01; ns—non-significant.

**Figure 4 pharmaceutics-17-01058-f004:**
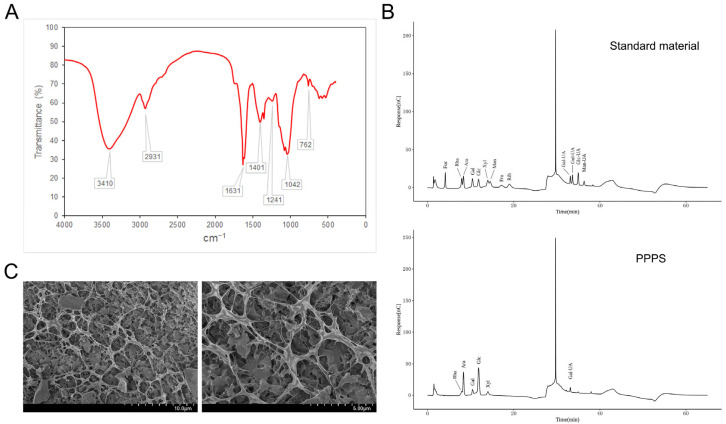
Structural characterization of PPPS. (**A**) FTIR detection. (**B**) Monosaccharide composition. (**C**) SEM detection.

**Figure 5 pharmaceutics-17-01058-f005:**
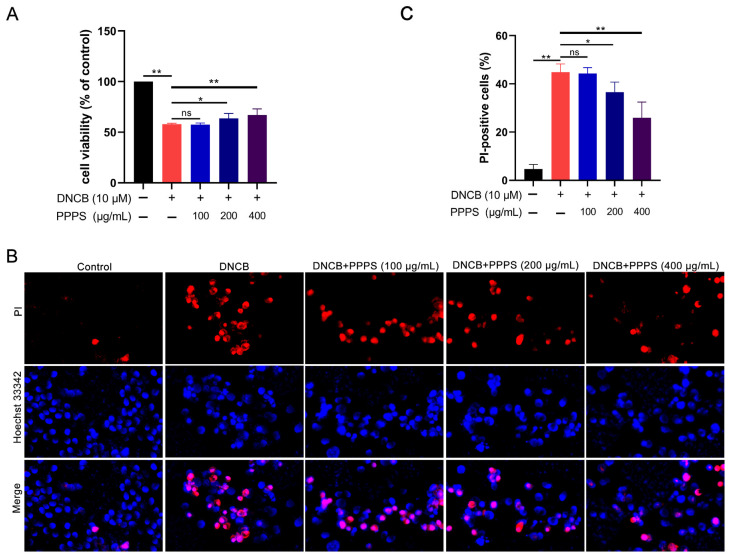
Therapeutic effects of PPPS on DNCB-damaged HaCaT cells. (**A**) PPPS dose–response on cell viability; (**B**) representative fluorescence images; (**C**) quantitative analysis of cells positive for PI. * *p* < 0.05; ** *p* < 0.01; ns—non-significant.

**Figure 6 pharmaceutics-17-01058-f006:**
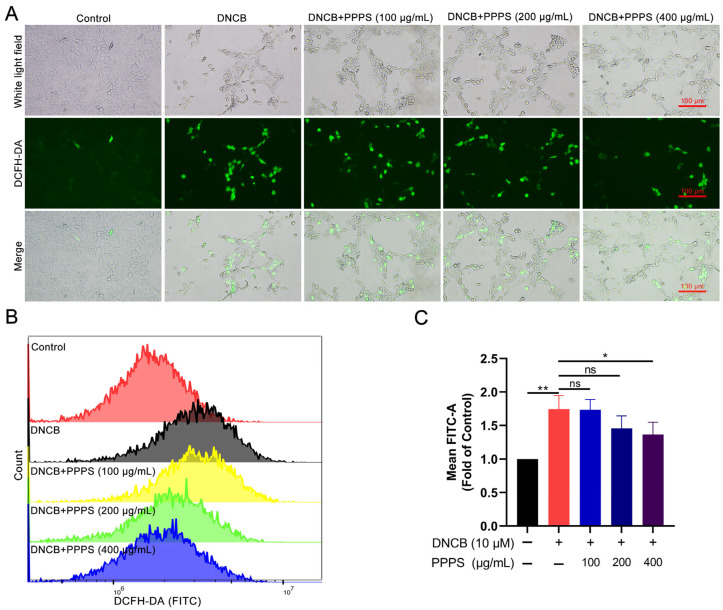
PPPS modulate intracellular ROS levels in DNCB-stimulated HaCaT cells. (**A**) Representative fluorescence images (green represents ROS-positive cells). (**B**) Flow cytometric quantification of ROS levels. (**C**) Corresponding quantitative analysis. * *p* < 0.05; ** *p* < 0.01; ns—non-significant.

**Figure 7 pharmaceutics-17-01058-f007:**
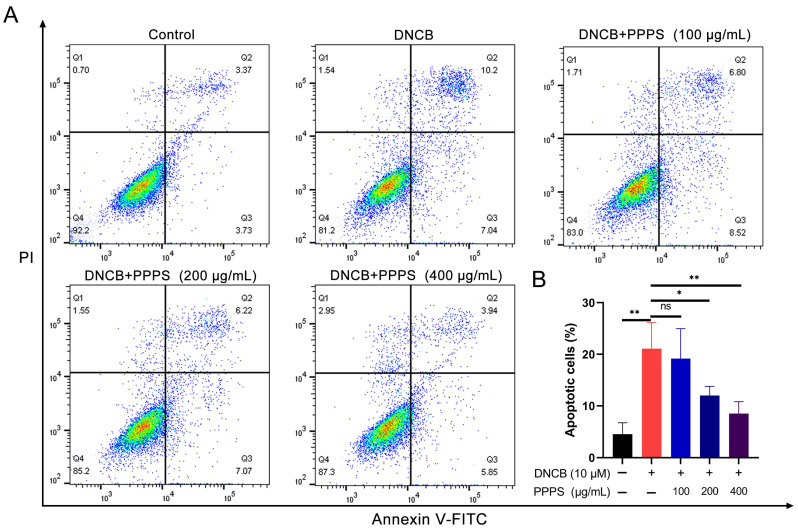
PPPS suppress DNCB-induced apoptosis in HaCaT cells. (**A**) Annexin V-FITC (x-axis) vs. PI fluorescence (y-axis). Quadrants: Q1 (mechanically damaged), Q2 (early apoptotic), Q3 (late apoptotic), and Q4 (viable). (**B**) Measurement of total apoptotic cells (Q2 + Q3). * *p* < 0.05; ** *p* < 0.01; ns—non-significant.

**Figure 8 pharmaceutics-17-01058-f008:**
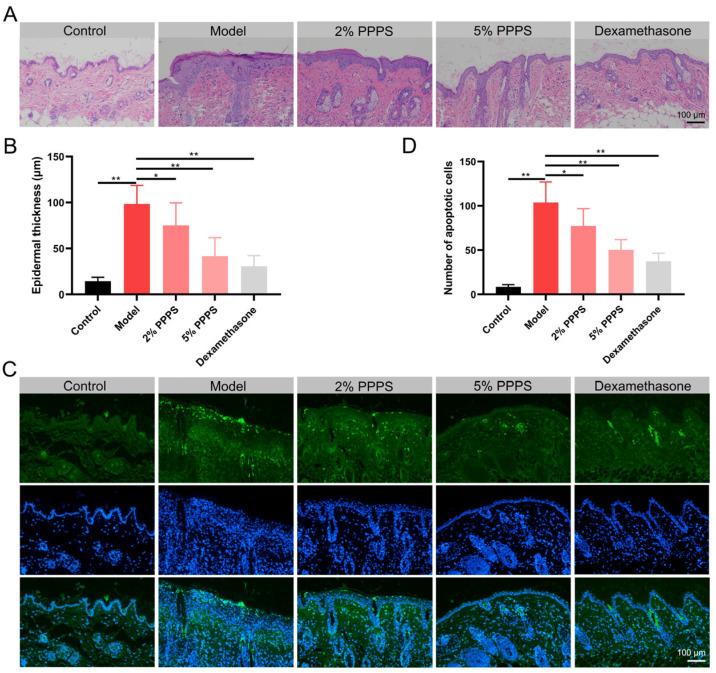
Therapeutic effects of PPPS on skin lesions in AD. (**A**) H&E-stained dorsal skin sections. (**B**) Epidermal thickness. (**C**) TUNEL-stained apoptotic cells (green) and cell nuclei (blue). (**D**) Counting of apoptotic cells. * *p* < 0.05; ** *p* < 0.01.

**Figure 9 pharmaceutics-17-01058-f009:**
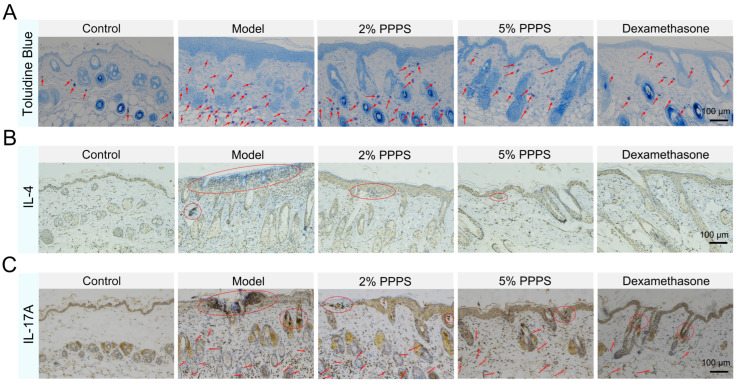
Effects of PPPS on immune cell-related factor expression in AD. (**A**) Toluidine blue staining of dorsal skin tissues (MCs indicated by blue granules, representative positive cells are indicated by arrows). (**B**) IHC analysis of IL-4 expression (representative positive areas are indicated by circles). (**C**) IHC analysis of IL-17A expression (representative positive areas are indicated by arrows and circles).

## Data Availability

Data will be made available on request.

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
