# Peer review of "A Sponge-like Polysaccharide from Pine Pollen: Structural Features and Therapeutic Potential in DNCB-Induced Atopic Dermatitis Models"

_pharmaceutics, 2025, doi:10.3390/pharmaceutics17081058_

Round 1
Reviewer 1 Report
Comments and Suggestions for Authors
Manuscript ”A Sponge-like Polysaccharide from Pine Pollen: Structural Features and Bioactivity Against Atopic Dermatitis” describes effect of pine pollen components on cellular and animal model of AD. The study seems carefully conducted however have major drawback which needs to be addressed to reasonable extent.
- PPPS as a material is not sufficiently described. PPPS seems to be a polymer. What are the physicochemical characteristics of this compound/polymr? Molecular weight? If this is a particle, then what is the size? Does the polymer have some charge? Is this linear or branched polymer?
- Treatment with PPPS seems positively affects cell viability, reduces ROS, etc. The effects are compared when toxicity is induced by DNCB in the presence and in the absence of PPPS. The effects are concentration dependent. Does DNCB interact with PPPS directly, so that effective concentration of DNCB is reduced due to DNCB adsorption to or complexation with PPPS polymer?
- The mechanism of PPPS is completely not clear, which makes this study less interesting. Does PPPS need to penetrate cell walls or stratum corneum in order to exhibit the described effects? It is not the first time when pollen polysaccharides are pointed to exhibit positive effects against AD and other pathologies. Authors should review theses mechanisms and at least indicate what could be possible mechanistic hypothesis of PPPS action which are relevant for their study. Mechanistic discussion should be reflected in introduction as well as in the discussion part of the manuscript.
Minor issues:
- Line 72. Please correct DNCB full name.
- Line 141. DCFH-DA is not mentioned or spelled before in the text.
- Line 235 and in other places Fig. 4 is noted instead of Fig. 3.
- Line 247 states that “PPPS possesses excellent hydrophilicity, moisture retention capacity, and ….” These characteristics have not been studied.
- 3B. Standard material should be better described in terms of component ratios.
Author Response
Reply to Reviewer 1:
Thank you very much for your time involved in reviewing the manuscript and your very encouraging comments on the merits. We appreciate your clear and detailed feedback, the new submit manuscript was carefully revised based on your comments, and hope that the explanations and modifications have adequately addressed all your concerns. In the remainder of this letter, we discuss each of your comments individually along with our corresponding responses.
Major issues:
Comments1: PPPS as a material is not sufficiently described. PPPS seems to be a polymer. What are the physicochemical characteristics of this compound/polymr? Molecular weight? If this is a particle, then what is the size? Does the polymer have some charge? Is this linear or branched polymer?
Response1: Thank you for your valuable comments. According to the problems you pointed out, we make the corresponding supplement in line 93-97 of the manuscript, which is as follows:
A PPPS is a kind of polar, hydrophilic macromolecule, such as glucose, galactose, arabinose, glucuronic acid, and others, connected via a glycosidic bond of monosaccharide composition. PPPS are branched polymers. The molecular weight of PPPS, assessed using various extraction methods, ranges between 25 and 128 kDa [13].
Comments2: Treatment with PPPS seems positively affects cell viability, reduces ROS, etc. The effects are compared when toxicity is induced by DNCB in the presence and in the absence of PPPS. The effects are concentration dependent. Does DNCB interact with PPPS directly, so that effective concentration of DNCB is reduced due to DNCB adsorption to or complexation with PPPS polymer?
Response2: Thank you for your comment. In this study, we indeed did not investigate the interaction between PPPS and DNCB, which indicates the limitations of this research. However, based on the published literature, it is speculated that the possibility of their combination is very small. Considering issues such as time, equipment and funds, we have supplemented this part as the limitation of this research in the discussion section of the article. The content is as follows in line416-436:
While this study demonstrates that PPPS alleviate DNCB-induced AD-like dam-age, whether their therapeutic effect results from direct binding between PPPS and DNCB remains unclear. DNCB contains highly polar -NOâ‚‚ and -Cl groups and func-tions as a hapten, forming immunogenic complexes through covalent binding with epidermal proteins [58]. Conversely, PPPS are a type of polysaccharide polymer rich in hydroxyl (-OH) and carboxyl (-COOH) groups [59]. Chemically, the potential for direct PPPS–DNCB interaction appears limited due to the insufficient structural basis for strong intermolecular forces. As a hapten, DNCB primarily mediates effects through specific protein binding, while PPPS—composed of multiple monosaccharides linked via β-glycosidic bonds—exhibits molecular polar groups with a greater propensity to form hydrogen bonds with aqueous-phase water molecules than to preferentially in-teract with DNCB's nitro groups. Furthermore, in published polysaccharide interven-tion studies using DNCB-induced AD models [60-63], polysaccharide–DNCB binding remains uninvestigated, with therapeutic efficacy consistently being attributed to im-munomodulatory pathways or antioxidant activity rather than a reduction in bioac-tive DNCB concentrations. This collective evidence suggests that polysaccharides exert therapeutic effects primarily by suppressing DNCB-triggered immune activation ra-ther than through direct molecular interaction. Consequently, our findings more strongly support the fact that PPPS alleviate DNCB-induced AD-like symptoms through intrinsic bioactivity. Future studies employing surface plasmon resonance (SPR) and high-performance liquid chromatography (HPLC) could quantitatively as-sess potential interactions between polysaccharides and DNCB.
Comments3: The mechanism of PPPS is completely not clear, which makes this study less interesting. Does PPPS need to penetrate cell walls or stratum corneum in order to exhibit the described effects? It is not the first time when pollen polysaccharides are pointed to exhibit positive effects against AD and other pathologies. Authors should review theses mechanisms and at least indicate what could be possible mechanistic hypothesis of PPPS action which are relevant for their study. Mechanistic discussion should be reflected in introduction as well as in the discussion part of the manuscript.
Response3: Thank you for your suggestion. In this study, we investigated the therapeutic effects of pine pollen polysaccharides. As you mentioned, we did not conduct in-depth mechanism exploration. Based on your suggestion, we supplemented the possible mechanism hypotheses of pine pollen polysaccharides in the discussion. The content is as follows in line 397-415:
Plant polysaccharides demonstrate therapeutic efficacy against atopic dermatitis (AD), though their mechanisms remain incompletely characterized. Zhang et al. re-vealed that honeysuckle polysaccharide (WLJP-025p) exerts anti-AD effects through the modulation of the MAPK/NF-κB/AP-1 axis [51]. Similarly, Liao et al. confirmed that Dendrobium officinale polysaccharide ameliorates DNFB-induced murine AD by suppressing MAPK/NF-κB/STAT3 signaling [52]. Our preliminary research established that PPPS alleviate LPS-induced myocardial inflammation by upregulating p110β ex-pression and inhibiting PI3K/AKT/NF-κB activation [53]. This evidence collectively suggests the NF-κB pathway represents a pivotal mechanism for PPPS-mediated an-ti-AD activity. As the central transcriptional regulator for multiple pro-inflammatory genes, NF-κB critically governs immune and inflammatory responses [54], with its ca-nonical pathway being crucial for effector T-cell differentiation and memory T-cell re-call responses [55]. The disruption of canonical NF-κB signaling consequently impedes Th2/Th17 cell differentiation, downregulating the transcription of pro-inflammatory mediators including TNF-α, IL-17A, and IL-4 [56, 57], thereby attenuating keratinocyte damage and immune dysregulation. Experimentally, PPPS significantly inhibited the DNCB-induced overexpression of IL-4 and IL-17A. Concurrently, its antioxidant properties likely indirectly dampen oxidative stress-mediated NF-κB activation, syn-ergistically inhibiting inflammatory cascades and providing crucial mechanistic in-sights for the PPPS-mediated attenuation of AD-like symptoms.
Given that DNCB-induced AD manifestations primarily feature epidermal barrier disruption, we propose that topical PPPS application alleviates these symptoms through direct local absorption, and we gratefully acknowledge your valuable recommendations for refining this perspective.
Minor issues:
Comments1: Line 72. Please correct DNCB full name.
Response1: Thank you for your careful examination, and we are sorry for our carelessness. In view of this problem, we have made corrections in the revised draft. As follows:
In line 79, "4-dinitrochlorobenzene" is corrected to "2,4-dinitrochlorobenzene".
Comments2: Line 141. DCFH-DA is not mentioned or spelled before in the text.
Response2: Thank you for your comments. In response to your suggestions, we have made the relevant supplements in the revised manuscript. Details are as follows:
In line 158, "DCFH-DA" has been corrected to "2,7-Dichlorodihydrofluorescein diacetate (DCFH-DA)".
Comments3: Line 235 and in other places Fig. 4 is noted instead of Fig. 3.
Response3: Thank you for your careful review. We apologize for our carelessness. Regarding this issue, we have corrected the revised manuscript.
Comments4: Line 247 states that “PPPS possesses excellent hydrophilicity, moisture retention capacity, and ….” These characteristics have not been studied.
Response4: Thank you for your suggestions on improving the rigor of the manuscript. Following your reminder, we have revised the sentence in the manuscript (line 268-269). The original sentence "PPPS possesses excellent hydrophilicity, moisture retention capacity, and …" has been modified to "PPPS may have excellent moisture retention capacity and biocompatibility."
Comments5: 3B. Standard material should be better described in terms of component ratios.
Response5: Thank you for your comments. In response to your suggestions, we have supplemented the information regarding the component ratio of PPPS in the revised manuscript as follows:
In Section 3.3, line 263-266, we have added: In Figure 4B, the results regarding monosaccharide composition show that PPPS were heteropolysaccharides composed of Rha (6.33%), Ara (27.01%), Gal (6.92%), Glc (45.72%), Xyl (5.83%), and Gal-UA (8.19%).
Thank you very much for your time involved in reviewing the manuscript. Based on your comments, we have replied and revised the manuscript. We hope this revised manuscript has addressed your concerns. Please contact us if you have any questions. Thanks again and best wishes to you.
Reviewer 2 Report
Comments and Suggestions for Authors
I enjoyed reading your article which contained some well executed investigations into the in vitro and in vivo effects of a polysaccharide derived from Pine cones. I have a number of observations which you may want to consider regarding the reporting of this work:
Title - I do not think it is appropriate to include the term Atopic dermatitis in your title, as what you describe are in vitro and in vivo models of some aspects of this complex condition.
Line 32-34 - I do not think this is a necessary line in the introduction; I would recommend just defining the importance of AD in terms of the prevalence.
Line 87 - I am a little confused how the 8 became 9; the initial analysis identified 8 constituents but you decided to include polysaccharides also. Maybe a little more elaboration as to why polysaccharides were added to the list - particularly as it transpires that they were the active component after all. Also, are the polysaccharides, which are presumably numerous and varied, considered one species or many?
Line 113 - 'Spectroscopy' not 'rpectroscopy'
Methods general - details of the microscope(s) and image capture technology used. Also the use of the ImageJ system; was it used for all cell quantification, and if so what were the details of its use in each instance?
Section 2.13 - Details of the antibodies used should be included here, supplier, concentrations, negative controls etc.
Line 258 - 'markedly' is not a good scientific word, probably best to just stick to 'significantly'.
Line 267-268 - 'ROS critical contributor to epidermal cell damage' - reference required
Figure 5B - These are quite unconvincing ROS results as presented. The differences in Mean FITC Intensity should be made clearer, perhaps by showing numbers or better images. Any differences are not clear.
Line 280 - Apoptosis reference should be included to support the statement
Figure 6 Legend - define LL, LR,UR and UL or better just change to Q1,Q2,Q3 and Q4. Are all the cells in Figure 6B actually apoptotic or apoptotic and necrotic combined?
Figures 4,6 and & - what do the error bars denote?
Line 135 - Need more information about Image J
Lines 312 and 315 - what your are describing are not 'AD' mice, these are mice treated with DCNB to create a model for some of the clinical and pathological aspects of AD.
Section 3.8 - you only detected Th2 and Th17 markers; why did you not check for Th1 and Treg effects? The effect of the polysaccharides could have been more generalised, suppressing all or more inflammation regulatory pathways.
Line 329 - primordial rather than primordially
Line 350-353 - Is there actually any evidence that polysaccharides can stimulate a/o enzymes or scavenge directly? This seems like speculation.
General comment - I would have preferred an additional investigation into the direct impact of the polysaccharides on DCNB; for example is it possible that there could be an interaction between these two species which limits or alters the effects of DCNB on the keratinocytes and/or mouse skin? Some of the results could be explained in the polysaccharides were able to bind to or in some way interact with the DCNB, thereby limiting it's interaction with cells and tissues? For example, incubating cells with PPPS alone to look for changes in gene expression, cytokine expression etc, might have been useful.
Author Response
Reply to Reviewer 2:
Thank you very much for your time involved in reviewing the manuscript and your very encouraging comments on the merits. We appreciate your clear and detailed feedback, the new submit manuscript was carefully revised based on your comments, and hope that the explanations and modifications have adequately addressed all your concerns. In the remainder of this letter, we discuss each of your comments individually along with our corresponding responses.
Comments 1: Title - I do not think it is appropriate to include the term Atopic dermatitis in your title, as what you describe are in vitro and in vivo models of some aspects of this complex condition.
Response 1: We sincerely appreciate your valuable comments. Building upon precedent studies—specifically, “Ameliorative effects of Wikstroemia trichotoma 95% EtOH extract on a mouse model of DNCB-induced atopic dermatitis” (https://doi.org/10.1016/j.jep.2024.118398), “Effect of Neferine on DNCB-Induced Atopic Dermatitis in HaCaT Cells and BALB/c Mice” (https://doi.org/10.3390/ijms22158237), and “Inhibitory Effect of Centella asiaticaExtract on DNCB-Induced Atopic Dermatitis in HaCaT Cells and BALB/c Mice” (https://doi.org/10.3390/nu12020411)—we have modified our manuscript title to “A Sponge-like Polysaccharide from Pine Pollen: Structural Features and Therapeutic Potential in DNCB-Induced Atopic Dermatitis Models”. Should you recommend an alternative title aligned with journal conventions, we remain open to further refinement based on your expert guidance. We reiterate our gratitude for your constructive critique, which significantly strengthened this work.
Comments 2: Line 32-34 - I do not think this is a necessary line in the introduction; I would recommend just defining the importance of AD in terms of the prevalence.
Response 2: We gratefully acknowledge your meticulous review and confirm that requested content deletions have been implemented in Section 1 of the manuscript; your constructive suggestion has been instrumental in strengthening this work.
Comments 3: Line 87 - I am a little confused how the 8 became 9; the initial analysis identified 8 constituents but you decided to include polysaccharides also. Maybe a little more elaboration as to why polysaccharides were added to the list - particularly as it transpires that they were the active component after all. Also, are the polysaccharides, which are presumably numerous and varied, considered one species or many?
Response 3: We express sincere appreciation for your insightful comments. The rationale for selecting polysaccharides as the active constituent stems from two primary considerations: Firstly, polysaccharides have been increasingly recognized as principal bioactive components in herbs and foods, playing crucial roles in disease treatment and prevention (https://doi.org/10.1016/j.carbpol.2017.12.009), with representative examples including lentinan, astragalus polysaccharide, and Ganoderma lucidum polysaccharide—all officially approved therapeutic agents. Secondly, prior research has established the therapeutic efficacy of PPPS in skin injury (https://doi.org/10.1016/j.reth.2025.06.009, https://doi.org/10.1016/j.ijbiomac.2022.04.210), complemented by its anti-inflammatory properties (https://doi.org/10.1016/j.ijbiomac.2024.136473), thereby warranting its inclusion in our experimental design. Per your recommendation, we have incorporated the following addition in Section 2.2 (line97-101):
Based on previous studies confirming the therapeutic effect of PPPS on skin injuries and considering their anti-inflammatory properties [15], we incorporated PPPS into this investigation. Consequently, this study examines the bioactivities of 9 pine pollen components, comprising the 8 previously characterized small molecules and PPPS.
In addition to your question " Are the polysaccharides considered one species or many?" In this study, the polysaccharide used was the polysaccharide extracted from pine pollen, which is regarded as a polysaccharide here. The unclear description of this part has also been revised in the new manuscript. We gratefully acknowledge your invaluable guidance throughout this revision process.
Comments 4: Line 113 - 'Spectroscopy' not 'rpectroscopy'.
Response 4: Thanks for your careful checks. We are sorry for our carelessness. The typographical error 'rpectroscopy' has been corrected to 'spectroscopy' in Line 125 of the revised manuscript. Your guidance in enhancing textual precision is sincerely appreciated.
Comments 5: Methods general - details of the microscope(s) and image capture technology used. Also the use of the ImageJ system; was it used for all cell quantification, and if so what were the details of its use in each instance?
Response 5: Thank you for your careful inspection, and we are sorry for this oversight. In 2.6. PI staining, 2.11. TUNEL staining, and 2.12. Mast cells staining in the revised manuscript, we have added details of the use of Image J. As follows:
Specifically, images were imported into ImageJ and converted to 8-bit format. Following thresholding and binarization, sequential processing with Fill Holes (to close intracellular gaps) and Watershed (to separate adherent cells) was applied. The minimum area threshold for the Analyze Particles function was determined by measuring a population of representative small cells. Cell counts were quantified using this size criterion. Manual verification was performed on 10% of randomly selected images (if discrepancies were >5%, manual counting was conducted using ImageJ’s Multi-Point tool).
Or:
Then, images were imported into ImageJ and converted to 8-bit format. The Multi-Point tool was used to count cells by clicking on individual cells.
Comments 6: Section 2.13 - Details of the antibodies used should be included here, supplier, concentrations, negative controls etc.
Response 6: According to your suggestion, we have added the details of the antibodies used in Section 2.13 (line 216) of the manuscript. The red font is the supplementary content, as follows:
The sections were then incubated overnight at 4°C with primary antibodies, including IL-4 (1:100, #AF5142, Affinity) and IL-17A (1:100, #DF6127, Affinity), and then the corresponding horseradish peroxidase (HRP)-conjugated secondary antibodies were applied and incubated at 37°C for 1 h.
Comments 7: Line 258 - 'markedly' is not a good scientific word, probably best to just stick to 'significantly'.
Response 7: We sincerely appreciate your constructive recommendation and have replaced the term "markedly" with "significantly" in the revised manuscript (line 280).
Comments 8: Line 267-268 - 'ROS critical contributor to epidermal cell damage' - reference required.
Response 8: We gratefully acknowledge your insightful suggestion for enhancing manuscript accessibility. In the revised version (line 289-290), we have supplemented the reference titled "Hydrogel dressing integrating FAK inhibition and ROS scavenging for mechano-chemical treatment of atopic dermatitis" (https://doi.org/10.1038/s41467-023-38209-x) as supporting evidence for the assertion that 'ROS serves as a critical contributor to epidermal cell damage'.
Comments 9: Figure 5B - These are quite unconvincing ROS results as presented. The differences in Mean FITC Intensity should be made clearer, perhaps by showing numbers or better images. Any differences are not clear.
Response 9: Thanks for your comments, we have changed the presentation of ROS images in the revised manuscript and added bar chart statistics for Mean-FITC.

Comments 10: Line 280 - Apoptosis reference should be included to support the statement.
Response 10: Thanks for your great suggestion on improving the accessibility of our manuscript. We have implemented your recommendation by supplementing the manuscript (line 303-304) with the following key references to substantiate keratinocyte apoptosis as a critical pathological mechanism in AD-related epidermal damage: "Magnoflorine from Coptis chinese has the potential to treat DNCB-induced Atopic dermatits by inhibiting apoptosis of keratinocyte " (https://doi.org/10.1016/j.bmc.2019.115093) and "Role of YAP-related T cell imbalance and epidermal keratinocyte dysfunction in the pathogenesis of atopic dermatitis" (https://doi.org/10.1016/j.jdermsci.2020.12.004). These additions provide mechanistic validation for our experimental observations regarding keratinocyte apoptotic pathways in AD pathology.
Comments 11: Figure 6 Legend - define LL, LR,UR and UL or better just change to Q1,Q2,Q3 and Q4. Are all the cells in Figure 6B actually apoptotic or apoptotic and necrotic combined?
Response 11: Thanks for your comments, in the revised manuscript we have modified the legend of Figure 6 in accordance with your comments, replacing LL, LR, UR and UL with Q1, Q2, Q3 and Q4. In addition, the statistical results in Figure 6B represent the number of cells with early and late apoptosis, which we have supplemented and modified in the figure legend.
Comments 12: Figures 4,6 and & - what do the error bars denote?
Response 12: Thank you for your comments. The error bars in our figures uniformly represent Mean ± SD (Standard Deviation). The horizontal lines between groups denote "comparisons of differences between the two groups at each end of the line," a presentation method adopted in numerous authoritative journals (Cell: https://doi.org/10.1016/j.cell.2025.06.002; Nature: https://doi.org/10.1038/s41586-025-09223-4, https://doi.org/10.1038/s41586-025-09207-4).
Comments 13: Line 135 - Need more information about Image J.
Response 13: Thank you for your comments. According to your suggestions, we have supplemented the details of the use of Image J as follows (line 147-154):
Specifically, images were imported into ImageJ and converted to 8-bit format. Following thresholding and binarization, sequential processing with Fill Holes (to close intracellular gaps) and Watershed (to separate adherent cells) was applied. The minimum area threshold for the Analyze Particles function was determined by measuring a population of representative small cells. Cell counts were quantified using this size criterion. Manual verification was performed on 10% of randomly selected images (if discrepancies were >5%, manual counting was conducted using ImageJ’s Multi-Point tool).
Comments 14: Lines 312 and 315 - what your are describing are not 'AD' mice, these are mice treated with DCNB to create a model for some of the clinical and pathological aspects of AD.
Response 14: Thank you for your comments. In the revised manuscript, "AD mice" has been universally replaced with "DNCB-induced AD-like mouse model". Thank you again for your careful examination.
Comments 15: Section 3.8 - you only detected Th2 and Th17 markers; why did you not check for Th1 and Treg effects? The effect of the polysaccharides could have been more generalised, suppressing all or more inflammation regulatory pathways.
Response 15: We sincerely appreciate your constructive suggestion. Published literature on Atopic dermatitis (https://doi.org/10.1016/j.jaci.2025.05.007) and Novel insights into atopic dermatitis (https://doi.org/10.1016/j.jaci.2022.10.023) establishes Th2/Th17 as pivotal contributors to AD pathogenesis. Th2 cells participate in responses against extracellular pathogens yet simultaneously associate with allergic reactions, with their core effector cytokines IL-4 and IL-13 mediating barrier dysfunction and immune activation in AD. Type 2 immune activation in skin downregulates epidermal antimicrobial peptides and filaggrin expression, while key Th2 cytokines IL-4/IL-13 not only drive cutaneous inflammation but also trigger neuronal sensitization. Furthermore, research demonstrates that Th17-mediated cellular immunity promotes Th2 immune responses (https://doi.org/10.1007/s12016-025-09057-y). Nakajima S, et al. verified that IL-17A deficiency ameliorates hapten-induced AD-like skin lesions while reducing Th2 chemokine expression and IL-4-producing cell populations (https://doi.org/10.1038/jid.2014.51). Conversely, Th1 cells primarily mediate immune reactions against intracellular viral or bacterial pathogens through IFN-γ production, predominantly characterizing chronic AD phases. Transcriptomic analysis of African American AD patients corroborated Th2/Th17 as central drivers of cutaneous immune activation (https://doi.org/10.1038/s41598-021-90105-w). Additionally, emerging small-molecule therapeutics (non-antibody agents) undergoing clinical trials uniformly target inhibition of Th2/Th17 cytokines (Figure 1, https://doi.org/10.1016/j.jaci.2017.01.011). Consequently, we specifically focused detection of these two canonical pathogenic pathways (Th2 and Th17) as the central mediators of AD pathophysiology.

Figure 1 Targets of biologic agents for atopic dermatitis currently being evaluated in clinical trials
Comments 16: Line 329 - primordial rather than primordially.
Response 16: Thanks for your careful checks. In the revised manuscript (line 354), the term "primordially" has been replaced with "primordial".
Comments 17: Line 350-353 - Is there actually any evidence that polysaccharides can stimulate a/o enzymes or scavenge directly? This seems like speculation.
Response 17: Thank you for your comments. Accumulating evidence confirms that polysaccharides can influence enzyme activity and stability through either direct binding or indirect effects. Kagliwal et al. demonstrated that horseradish peroxidase (HRP) can bind to 10 different polysaccharides, and the HRP-polysaccharide conjugates exhibit enhanced stability and activity (https://doi.org/10.1016/j.ijbiomac.2014.05.065). Additionally, the article "Inflammatory caspases are innate immune receptors for intracellular LPS" published in Nature (https://doi.org/10.1038/nature13683) revealed that caspase-4/11 (a type of cysteine protease) directly and specifically binds to intracellular LPS (a polysaccharide extracted from the cell wall of Gram-negative bacteria) via its CARD domain, with the binding affinity being affected by the polysaccharide chain of LPS. Another study further confirmed the possibility of caspase-11 binding to specific domains of LPS (https://doi.org/10.1126/sciadv.adt9027).
Moreover, polysaccharides may affect enzyme activity through indirect mechanisms. Zhou et al. confirmed that hawthorn fruit polysaccharides reduced serum levels of alanine transaminase (ALT) and aspartate transaminase (AST), reversed the abnormal decrease in superoxide dismutase (SOD), and alleviated hepatic oxidative stress by restoring hepatic metabolic disorders (https://doi.org/10.1016/j.phymed.2025.156458).
Therefore, we propose that polysaccharides can influence enzyme activity to inhibit peroxidation, thereby alleviating the occurrence and progression of AD.
General comment - I would have preferred an additional investigation into the direct impact of the polysaccharides on DCNB; for example is it possible that there could be an interaction between these two species which limits or alters the effects of DCNB on the keratinocytes and/or mouse skin? Some of the results could be explained in the polysaccharides were able to bind to or in some way interact with the DCNB, thereby limiting it's interaction with cells and tissues? For example, incubating cells with PPPS alone to look for changes in gene expression, cytokine expression etc, might have been useful.
Response:
Thank you for your comment. In this study, we indeed did not investigate the interaction between PPPS and DNCB, which indicates the limitations of this research. However, based on the published literature, it is speculated that the possibility of their combination is very small. Considering issues such as time, equipment and funds, we have supplemented this part as the limitation of this research in the discussion section of the article. The content is as follows in line 416-436:
While this study demonstrates that PPPS alleviate DNCB-induced AD-like damage, whether their therapeutic effect results from direct binding between PPPS and DNCB remains unclear. DNCB contains highly polar -NOâ‚‚ and -Cl groups and functions as a hapten, forming immunogenic complexes through covalent binding with epidermal proteins [58]. Conversely, PPPS are a type of polysaccharide polymer rich in hydroxyl (-OH) and carboxyl (-COOH) groups [59]. Chemically, the potential for direct PPPS–DNCB interaction appears limited due to the insufficient structural basis for strong intermolecular forces. As a hapten, DNCB primarily mediates effects through specific protein binding, while PPPS—composed of multiple monosaccharides linked via β-glycosidic bonds—exhibits molecular polar groups with a greater propensity to form hydrogen bonds with aqueous-phase water molecules than to preferentially interact with DNCB's nitro groups. Furthermore, in published polysaccharide intervention studies using DNCB-induced AD models [60-63], polysaccharide–DNCB binding remains uninvestigated, with therapeutic efficacy consistently being attributed to immunomodulatory pathways or antioxidant activity rather than a reduction in bioactive DNCB concentrations. This collective evidence suggests that polysaccharides exert therapeutic effects primarily by suppressing DNCB-triggered immune activation rather than through direct molecular interaction. Consequently, our findings more strongly support the fact that PPPS alleviate DNCB-induced AD-like symptoms through intrinsic bioactivity. Future studies employing surface plasmon resonance (SPR) and high-performance liquid chromatography (HPLC) could quantitatively assess potential interactions between polysaccharides and DNCB.
Thank you very much for your time involved in reviewing the manuscript. Based on your comments, we have replied and revised the manuscript. We hope this revised manuscript has addressed your concerns. Please contact us if you have any questions. Thanks again and best wishes to you.
Reviewer 3 Report
Comments and Suggestions for Authors
- Lines 46–56 require support with suitable and up-to-date references to strengthen the scientific foundation.
- The section describing the isolation and identification of bioactive compounds should be expanded and presented in greater detail
- It is recommended to support the experimental procedures with relevant and recent references to validate the methodology.
- The inclusion of a graphical abstract is strongly recommended to visually summarize the key findings and enhance the manuscript’s appeal.
- The overall language quality is acceptable; however, minor revisions are suggested to enhance clarity, coherence, and flow across the manuscript.
Author Response
Reply to Reviewer 3:
Thank you very much for your time involved in reviewing the manuscript and your very encouraging comments on the merits. We appreciate your clear and detailed feedback, the new submit manuscript was carefully revised based on your comments, and hope that the explanations and modifications have adequately addressed all your concerns. Below this reply box, we have uploaded the Word file of Response, where we have discussed each of your comments one by one and attached our corresponding responses. We hope this revised manuscript has addressed your concerns. Please contact us if you have any questions. Thanks again and best wishes to you.
Comments 1: Lines 46–56 require support with suitable and up-to-date references to strengthen the scientific foundation.
Response 1: Thank you for your careful review. In the revised manuscript, we have updated and supplemented the references accordingly.
Comments 2: The section describing the isolation and identification of bioactive compounds should be expanded and presented in greater detail.
Response 2: Following the valuable suggestions received, the methodology for polysaccharide separation and extraction has been supplemented in Section 2.1, Line 67-75 of the Materials and Methods, as detailed below:
In total, 100 g of broken-wall pine pollen was defatted via reflux extraction with petroleum ether at 50°C for 3 h and subsequently extracted via reflux with 95% ethanol at 70°C for 2 h to remove impurities; the resulting material was then subjected to aqueous extraction by boiling it with 8-fold (w/w) water addition three times. After concentration of the combined filtrates to 200 mL, the polysaccharide solution was subjected to deproteinization with Sevage reagent (chloroform:butanol, 4:1 v/v) in a 3:1 volume ratio. The polysaccharide solution layer was collected, absolute ethanol was added to a final concentration of 80%, the precipitate was collected via centrifugation at 3000 × g for 10 min, and the final product (PPPS) was obtained through lyophilization.
Comments 3: It is recommended to support the experimental procedures with relevant and recent references to validate the methodology.
Response 3: Thank you for your careful review. In the revised manuscript, we have updated and supplemented the references accordingly.
Comments 4: The inclusion of a graphical abstract is strongly recommended to visually summarize the key findings and enhance the manuscript’s appeal.
Response 4: Thank you for your comments. Following your suggestions, we have added a graphical abstract, and the detailed content is included in the compressed file of the Figure.

Comments on the Quality of English Language:The overall language quality is acceptable; however, minor revisions are suggested to enhance clarity, coherence, and flow across the manuscript.
Response: Thank you for your comment regarding the language quality of the manuscript. We fully agree with this suggestion and have now utilized MDPI's English editing service to enhance the manuscript. The text has been carefully revised throughout to improve clarity, coherence, and overall readability. We believe these refinements meet the journal's linguistic standards.

Thank you very much for your time involved in reviewing the manuscript. Based on your comments, we have replied and revised the manuscript. We hope this revised manuscript has addressed your concerns. Please contact us if you have any questions. Thanks again and best wishes to you.